# What does good care look like to people living with congenital heart disease in the 21st century? Qualitative online, asynchronous discussion forums

Jo Wray [1,2] Christina Pagel [3] Louise Coats [4,5] Adrian H Chester,[6] Fiona Kennedy,[7] Sonya Crowe [3]

¹Heart and Lung Directorate, Great Ormond Street Hospital for Children NHS Foundation Trust, London, UK
²Institute of Cardiovascular Science, University College London, London, UK
³Clinical Operational Research Unit, Department of Mathematics, University College London, London, UK
⁴Freeman Hospital, Newcastle upon Tyne, UK
⁵Newcastle University, Newcastle upon Tyne, UK
⁶Heart Science Centre, Harefield, UK
⁷Barts Health NHS Trust, London, UK

**Correspondence to**
Dr Jo Wray;
Jo.Wray@gosh.nhs.uk

## ABSTRACT

**Objectives** As part of a wider study, our aim was to elicit perspectives of people with congenital heart disease (CHD) and/or their parents/carers about their experiences of healthcare and what is important to them when receiving care.

**Design and setting** A qualitative study involving a series of closed, asynchronous, online discussion forums underpinned by an interpretivist framework and set up and moderated by three patient charities via their Facebook pages.

**Participants** People with CHD and parents/carers of people with CHD from the UK.

**Results** Five forums were run for 12–24 weeks across the three charities, and 343 participants signed up to the forums. Four linked themes related to processes of care were identified following thematic analysis of the transcripts: relationships and communication; access and coordination; experience of discrete episodes of care and psychological support. These impacted how care was experienced and, for some patients, outcomes of CHD and its treatment as well as broader health outcomes. In addition, context relating to stages of the patient journey was described, together with patient-related factors such as patients' knowledge and expertise in their own condition.

**Conclusions** People with CHD and their parents/carers want individualised, person-centred care delivered within an appropriately resourced, multidisciplinary service. Although examples of excellent care were provided it is evident that, from the perspective of patients and parents/carers, some National Health Service Standards for people with CHD were not being met.

## STRENGTHS AND LIMITATIONS OF THIS STUDY

⇒ Data collection methods (asynchronous discussion forums on Facebook) facilitated accessibility for participants and resulted in a large sample of geographically and clinically diverse participants of different ages.
⇒ Use of social media to collect data supports participation by those who might otherwise not engage in research.
⇒ Most participants were female and ethnic diversity was limited.
⇒ The study design precluded participation from those who could not read English, use social media or were unfamiliar with, or without access to, appropriate technology.

## BACKGROUND

Congenital heart disease (CHD) is the most common congenital anomaly, with a global incidence of 6–8 per 1000 live births.[1] Successive improvements in surgical and medical care mean many more children with CHD now reach adulthood and live longer adult lives.[2,3] Furthermore, increasing numbers live with significant physical and mental health comorbidities alongside the natural progression of their CHD.[4–7] The vast majority of this heterogeneous group in the UK and other developed countries are advised to remain under specialist lifelong follow-up[8,9] though sporadic clinic non-attendance is frequent, particularly in adulthood,[10] and patients may be lost to follow-up entirely, sometimes with significant consequences.[11] Unlike many other chronic conditions, organisation of care for these patients remains, for the vast majority, under the remit of the specialist team at the tertiary hospital, which may be far from where they live. In general, routine investigations are performed annually, sometimes more frequently in those with complex or advanced disease and may include unanticipated invasive procedures.[10] Some may be seen in outreach clinics, usually by members of the specialist team but also by non-CHD cardiovascular specialists,[12] following a 'hub-and-spoke' model of delivering care, but these services remain underdeveloped.[13] Attendances in primary care and at Accident and Emergency (A&E), in similar care structures, are higher than for the general population.[14] Occasionally, people with CHD remain

undiagnosed until adulthood[15 16] when they typically present with symptoms.

The way in which patients (and their carers) experience care has a significant impact on their engagement with that care and the professionals delivering it.[17] In the National Health Service (NHS) Outcomes Framework[18] 'ensuring that people have a positive experience of care' was identified as a priority, recognising the consequences for safety and effectiveness. However, if the NHS is to provide optimal care that meets the needs and expectations of service users, it is important to understand what characterises 'good' care from the patient and their family's perspective.

In this study, we elicited the perspectives of patients and carers about their experiences of CHD care and how that related to how patients and carers would like care to be delivered. This was carried out as part of a wider study (Congenital Heart Audit: Measuring Progress In Outcomes Nationally)[19] aiming to develop tools for routine measurement of outcomes in CHD that are considered relevant to all stakeholders.

## METHODS
### Design

A qualitative approach underpinned by an interpretivist framework was used, in which participant (patient or parent/carer) views were elicited through online discussion forums.

### Patient and public involvement

A co-researcher who was also a CHD patient (AHC) led a patient and public involvement (PPI) group for the overall study (comprising three adults with CHD and one grandparent of a child with CHD). The PPI group reviewed and provided feedback on the content and language of questions for the online forum and on the findings prior to submission. Forum questions and presentation of the findings were revised based on feedback from the PPI group.

### Participants and data collection

We adopted an established approach.[20–22] Three UK CHD charities (Little Hearts Matter; Children's Heart Federation; Somerville Foundation) facilitated and moderated one or more closed, asynchronous, online discussion groups via their Facebook pages. This enabled us to elicit a range of views (parents/carers of younger (<12 years) children, adolescents and adults; adolescents and adults with CHD) across the spectrum of CHD severity (complex (single ventricle conditions)/less complex (biventricular conditions)). The research team developed questions and revised content and language based on feedback from the PPI group and charity collaborators. On the advice of the charities, separate forums were run for adults with CHD, adolescents with CHD and parents/carers of children and adolescents with CHD. Carers of adult patients participated in the adult discussion groups. Discussion

---

**Box 1  Examples of questions related to experiences of care for the adult patient forums**

Questions for the parent/carer and teenager forums were similar, with minor wording changes to reflect those respondent groups (eg, designed to appeal to teenagers or wording appropriate for carers rather than patients). Charities could introduce questions to probe further—for example, can you tell us more about a clinic appointment that went well or less well?

1. Understanding how patients and families judge how 'good' a service is will help us to identify things that services do well and those aspects where the National Health Service (NHS) could improve.
   – What do you think makes a service good quality? What sort of things do you want to know about a hospital or service to judge whether it is a good place to care for you?
   – What things might make you worry that the hospital/service was not as good quality as other hospitals/services?
2. We would like to know about what information you think should be collected about people's experience of the care they receive—for example, about how health professionals talk to patients and their families.
   – Thinking about some recent care you have had, are there any particular aspects of your experience that the NHS could collect about those services to judge their quality?
3. We would like to ask you about your experience of attending clinic appointments, which we recognise is not always easy for patients. How easy or difficult have you found it to keep up with follow-up appointments? Are there particular things that make it easy or difficult for you to attend follow-up appointments? How easy do you find it to see professionals other than doctors and nurses when you attend clinic, if you want to (such as social workers, psychologists, play specialists)?
4. We'd like to think now about some particular things that may happen to a patient with CHD. We know that patients with CHD often have to undergo a number of medical interventions, such as catheters and heart surgery. We are interested in understanding how patients and families judge whether an operation or catheter has gone well as this will help us to identify areas where hospitals can improve the service they provide.
   – Have you had a medical intervention for CHD (surgery or a catheter procedure) since you have been in adult services? If so, thinking back to that time, can you remember what was most important to you about the intervention and the care you received?
   – Making the decision to agree to an operation or catheter can be a big step. Do you think you were given enough information to make the decision? Do you remember what you were told about the likely outcomes of the procedure and how it might affect you?
   – And what about now? Thinking about that medical intervention, are there other things that are important to you now about how it went and any impact it has had? Were your expectations about your surgery met, both at the time and now?

---

forums were advertised by each charity on their web page with potential participants signposted to the charity's Facebook page for further information about the study. People wanting to participate were asked to provide basic demographic details and subsequently directed to the appropriate closed Facebook group where they could join the discussion. Enrolment was ongoing, such that new participants could join at any stage while the forum was running; they could see and respond to questions

**Table 1** Characteristics of those who signed up to the online forums (n=343)

| | Number (%) |
|---|---|
| Participants: adults with CHD | 235 (69) |
| Young people with CHD | 11 (3) |
| Parents/carers of adult patients with CHD | 7 (2) |
| Parents/carers of children with CHD | 90 (26) |
| Participant gender: male | 33 (10) |
| Female | 245 (71) |
| Unknown | 65 (19) |
| Participant age group: <16 years | 2 (1) |
| 16–20 | 11 (3) |
| 21–30 | 32 (9) |
| 31–40 | 69 (20) |
| 41–50 | 72 (21) |
| 51–60 | 65 (19) |
| 61+ years | 15 (4) |
| Unknown | 77 (22) |
| Age group of person with CHD: 0–1 years | 9 (3) |
| 2–5 years | 17 (4) |
| 6–10 years | 15 (4) |
| 11–15 years | 8 (2) |
| 16–18 years | 9 (3) |
| 19+ years | 238 (69) |
| Unknown | 47 (14) |
| Participant ethnicity: white | 326 (95) |
| Non-white | 5 (1) |
| Unknown | 12 (3) |
| Location of specialist service: England (North East) | 11 (3) |
| England (North West) | 26 (6) |
| England (Yorkshire and the Humber) | 20 (6) |
| England (East Midlands) | 14 (4) |
| England (West Midlands) | 47 (14) |
| England (East of England) | 9 (3) |
| England (London) | 75 (22) |
| England (South East) | 28 (8) |
| England (South West) | 27 (8) |
| Wales | 8 (2) |
| Scotland | 24 (7) |
| Northern Ireland/other | 15 (4) |
| Unknown/none | 39 (11) |
| Location of home: England (North East) | 13 (4) |
| England (North West) | 43 (13) |
| England (Yorkshire and the Humber) | 22 (6) |
| England (East Midlands) | 20 (6) |
| England (West Midlands) | 36 (10) |

Continued

**Table 1** Continued

| | Number (%) |
|---|---|
| England (East of England) | 23 (7) |
| England (London) | 22 (6) |
| England (South East) | 57 (17) |
| England (South West) | 46 (13) |
| Wales | 21 (6) |
| Scotland | 25 (7) |
| Northern Ireland/other | 8 (2) |
| Unknown | 7 (2) |
| Complexity of CHD: single ventricle condition | 121 (35) |
| Biventricular condition | 216 (63) |
| Unknown | 6 (2) |

A number of participants chose not to provide some or any demographic information.
CHD, congenital heart disease.

that had already been posted if they wished. Charities could introduce prompts or post new questions, based on responses. Questions for each forum were similar, with minor wording revisions to reflect the respondent group (box 1). Forums ran for 12–24 weeks (during February–August 2021, dependent on responses and interruptions due to COVID-19). Charities were responsible for all day-to-day running and moderation of forums in line with a standard operating procedure (SOP) previously developed in collaboration with the research team.[21] The SOP included processes for managing inappropriate and/or offensive messaging and distressed users as well as procedures for the day-to-day running of the forum.

### Data management and analysis
Identifiable data were removed by the charities before the transfer of the anonymised transcript with aggregated demographics to the research team. Transcripts were thematically analysed[23] independently by three research team members (JW, FK and LC): codes were given to segments of data and similar codes were grouped to create themes related to the experience of CHD care. The researchers met to discuss and finalise themes. Themes and transcripts were sent to an additional research team member (not involved in coding) to review (SC) to ensure all data about experience of care were accurately represented in the themes.

### RESULTS
Five forums (two for adults with CHD, two for parents/carers and one for younger patients (<18 years of age)) were run across three charities. In total, 343 participants signed up to the forums. Demographics of participants are shown in table 1.

Four linked themes related to processes of care were identified: relationships and communication, access and

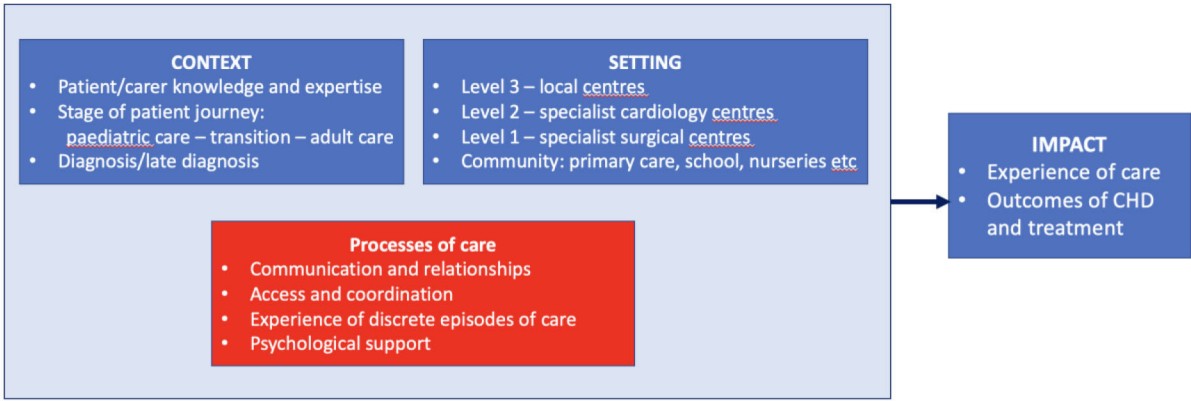

**Figure 1** Map of the themes developed from the data. CHD, congenital heart disease.

coordination, experience of discrete episodes of care and psychological support. These impacted how care was experienced and, for some patients, outcomes of CHD and its treatment as well as broader health outcomes. Processes of care were described in different cardiac and community settings. In addition, context relating to stages of the patient journey, time of diagnosis and patient-related factors, for example patients' expertise in their own condition, were described (figure 1). Themes are described below, with illustrative quotes from adults with CHD unless otherwise stated.

### Relationships and communication

The concepts of relationships and communication were intertwined and evident throughout all forum groups. Participants discussed the importance of the patient–clinician relationship in terms of the characteristics of the clinician and the relationship. People with CHD want to have a '*great relationship with [the] consultant, knowing you'll see someone who 'knows' you and your condition really well',* and emphasised the need for awareness of, and continuity in, those providing care: '*They don't have to be my mate but it's important that they're cognisant of the context because CHD is lifelong and affects all parts of my life.'* The importance of professionals being kind, approachable, honest and caring as well as being excellent communicators and expert in CHD were highlighted. Participants felt they needed a team they respect and trust, who listened and treated them '*like a human being and not a burden on resources. The impact of being told that—both overtly and implicitly—is severe, and affects engagement.'* A parent of a child said of the surgeon: '*His calm, kind, confident communication and how he took personal responsibility was pivotal in giving me the confidence to trust him to operate on my son.'* Another parent said*: 'The best care we have received has been when we were listened to and not judged for the questions and requirements we had.'* Adolescents with CHD commented on the value of clinicians being interested in them beyond their heart condition: '*When I was in children's I loved that my cardiologist took an interest in my life and asked about hobbies, school and social life. I feel if more people did this you would feel even more welcomed and calm and not nervous.'* Other patients described the importance of being able to '*maintain a great relationship with the*

*CNS* [clinical nurse specialist] *nurses as often it is hard to remember or say everything needed to the consultant,'* with the need for others (staff and/or family members) to advocate for them when they feel unable to do so also acknowledged. Recognition that adults with CHD are experts in their own care and need to feel empowered and believed was highlighted, with emphasis on the importance of feeling part of the team: '*I like the feeling that my medical care is a partnership between the cardiac team (who know about hearts) and me (I know about me and what's normal/ok/priority for me).'* Another commented: '*I hope the team understands a little of what I've been through, what I bring with me in terms of experience, knowledge but also emotionally.'* Participants also discussed the importance of being involved in decision-making, '*What's important to me is that all my team totally understand my condition, listen to me and include me in any decisions'.* One participant talked about being able to make '*Self-referrals, avoiding local cardiac care, we know when we feel shitty'.* Some participants also discussed occasions when they had been admitted to local hospitals where they were not known and communication was poor, with consequences for broader health outcomes: '*My local district general hospital also scared the pants off me after being admitted with palpitations when a cardiologist with 'an interest in GUCH' decided to hint on my life expectancy…something I'd never asked for and which has damaged my mental health ever since'.*

Practical aspects of communication with professionals were discussed in terms of functional elements (eg, timeliness), content (eg, information about procedures and associated risks, accuracy of content) and the mechanisms of communication: '*…results from tests being given by consultant not just a short letter that is sent to GP'.* Decision-making was also discussed in relation to the importance of being given honest feedback: '*I'd rather have the detail with an explanation about what this means even though it might be difficult to come to terms with. I want to be making informed decisions.'*

Many people with CHD are under the care of multiple teams and specialists and participants described the importance (and often failure) of communication between professionals to ensure joined-up care—'*Could*

that all have been avoided if the surgical team had involved cardiology more? Better communication certainly would have helped either way'—and discussed communication between teams—'why do the ACHD team and the EP team not seem to talk to one another? What's the point of coming up with a plan with the one team and then the other walking in and saying something different?' They also described challenges in communication between specialist centres and outreach and primary care services. As one participant said, 'I'm seen at an outreach clinic and there is quite often a lack of communication between the outreach clinic and the main hospital' while another described the importance of there being 'a note on the GP system so that when you appear with heart symptoms they take it seriously'. In contrast, a mother of a paediatric patient commented: 'I feel very privileged to have such a joined-up way of all the hospitals and primary care teams working together for my son's health'.

Parents of paediatric patients discussed the importance of communication with schools and nurseries and getting help with education, health and care plans: 'Schools always panic over everything. My son's first week back at school I was called 3 times by the matron due to little things. I asked cardiac nurses for info for school and was just given general information leaflets. They should be more person-centred as 2 kids with the same condition will be at different stages and have different symptoms.'

### Access and coordination

Respondents described challenges accessing care when they were moving to another hospital, out of hours or when they were unwell or needed advice: 'Knowing that I can ask a question or get a query answered quickly is very important to me. So, timely and accessible care is important'. Some people with CHD had been lost to follow-up, sometimes with a significant impact on their health outcomes: 'my checkups were not continued…notes must have been lost somewhere. I had no drugs, nothing and I had a stroke which I believe could have been avoided' or had to fight to get treatment: 'Actually getting any care at all would be good… I haven't been referred to the ACHD team despite asking and my 'care' has been non-existent'. Others talked about 'being lost in the system…never followed-up' or a lack of appropriate action when they were seen: 'Saw him [consultant] once then nothing happened. No blood thinner etc. I had a stroke and am so bitter…my life is in tatters due to lack of minimal care'. In contrast, some parents talked about the ready access they had to the clinical team or an open-door policy to access the ward if their child was unwell: 'My son is still looked after [by] a multidisciplinary team over two hospitals and we have 24 hour access to a local hospital that also have wonderful and caring doctors. My GP called us at the various hospitals to check we were ok.'

Coordination of care was mentioned frequently, particularly in relation to appointments and investigations and organising them to happen at the same time ('Appointments in a cluster—MRI, echo etc so we don't have multiple trips to manage'), with some highlighting examples of good practice: 'In recent years (pre-Covid) it has gone

like clockwork and I'm usually in hospital for a couple of hours at most.' Others felt that changes to clinic arrangements made during COVID-19 were positive and should be continued: 'What has been useful in Covid times is going for tests and then having a video or telephone conversation to discuss those test results. I think this is a good model to take forward.' Others talked about the complexity of their problems and needing everything to be together: 'I would like somewhere to go where all our older problems are taken care of…heart, everything under one roof, not non-guch [ACHD]* hosp where docs don't understand your condition'. Another respondent said: 'Definitely not having consultant appointment- test arranged for 2 months later, another test a couple of months after that and because of that result another test after that with a consultation several months after that which takes you a year down the line to then have a consultation with someone who hasn't even looked at the results of the tests!' (*Patients often refer to GUCH, which stands for grown-up congenital heart disease, now referred to as ACHD or adult congenital heart disease).

### Experience of discrete episodes of care

Participants discussed experiences of clinics and inpatient stays, describing the importance of being given enough time ('appointments not to be rushed'), being made to feel 'at home…and calm' (adolescent), their experiences of undergoing tests and the people conducting them, and what makes attending difficult. Experience of inpatient stays was described in terms of the environment, people working on the ward (particularly in relation to staff changeover at the end of a shift and whether staff had read their medical notes: 'We find every time my son is admitted to hospital the staff do not seem to read his notes and start to panic that he has an abnormal heart rhythm. Also seeing different doctors each day and them saying different things'), ability to sleep on the ward, available activities (children) and experience of discharge: 'Having a good discharge! Why oh why can it take hours and hours to get out of hospital when you are clinically well enough to leave. Getting out of hospital is as important as being admitted in the first place.' Respondents highlighted the value of being routinely asked about their experiences of outpatient and inpatient care through the completion of patient-reported experience measures and the importance of telling patients what has been or will be done with the information. Linked to this, some adults with CHD articulated the importance of patients being involved in service development initiatives.

### Psychological support

Mental health and the need for psychological support were mentioned as important but often lacking. Some described waiting for psychology support for many years or services not being available at all: 'Access to psychology at clinics—that is non-existent', a scenario also reported in paediatric services: 'Lack of ongoing psychology support—we have waited 4 years' (parent). Some adults received a late diagnosis, with repercussions for their mental health: 'I found out later in life and I found the mental health support was limited, specifically with adjusting to the news that my heart

*could have ripped itself apart. Those who are diagnosed later or suddenly escalate should be referred to mental health specialists as a priority.'* Another participant who had experienced psychological issues for which they had not received any support commented: *'My story exposed the importance of psychological care and the potential physical damage that can be caused to the cardiological system from stress, anxiety and childhood trauma instigated by CHD.'* The importance of other professionals in the wider team was also identified, including for children—*'I think that every child that had life-long illness should have regular play therapy, to help deal with their fear and emotions they face.'* Finally, the valuable role of patient support organisations was acknowledged, *'They put me in touch with [charity], who have been my saviour.'*

## DISCUSSION

In this study, our aim was to understand the perspectives of people with CHD and their parents/carers of their experiences of care and how that related to the ways in which they would like care to be delivered and, as part of a wider project, outcomes evaluated. Four key themes were identified (relationships and communication, access and coordination, experiences of discrete episodes of care and psychological support) with participants providing examples of when things went well and less well. The findings support those of previous research with people with CHD[24–29] as well as other groups of people with chronic illness,[30–32] in which the importance of care being delivered by competent clinicians, effective patient–clinician communication, coordination of and access to care and the need for psychological support have all been highlighted. However, of concern, is that the findings suggest that a number of the NHS CHD recommended standards and service specifications[33] are presently not being met, for example, psychology provision

and referral to ACHD services—although there are examples of excellent care, particularly in paediatric services. Furthermore, the findings indicate that, contrary to the NHS vision of promoting equality and reducing health inequities, service delivery is variable between centres in the UK.

There was a degree of overlap between themes, particularly between the themes of 'relationships and communication' and 'access and coordination'. The importance of relationships between patients and clinicians is well documented.[29 34 35] People with CHD often have complex diagnoses and patients highlighted how they wanted professionals to really know about their specific diagnosis and treatment pathway. Communication is part of this, not only between patients and professionals but also between different professionals as these patients are often under multiple specialities or seen in several centres. This also links to coordination of care and the need for a more joined-up approach. The majority of our participants were adult patients and much of what they reported supports the view that the current way in which ACHD ambulatory care is delivered in the UK (the hub and spoke approach)[13] does not provide either equitable or patient-centred care.[36] Poor communication and coordination of care also resulted in some people with CHD being lost to follow-up, particularly at times of transition from paediatric to adult services or from one hospital to another or suffering potentially avoidable complications, with consequences for longer-term health outcomes. While these findings are not new, the complexity of this patient cohort and the requirement for lifelong follow-up increase their vulnerability to poorer outcomes and experiences.

Building on the conceptual map of Entwistle *et al*[37] who undertook a critical interpretive synthesis of research literature on patients' perspectives of healthcare delivery, we have used our findings to characterise what 'good care' looks

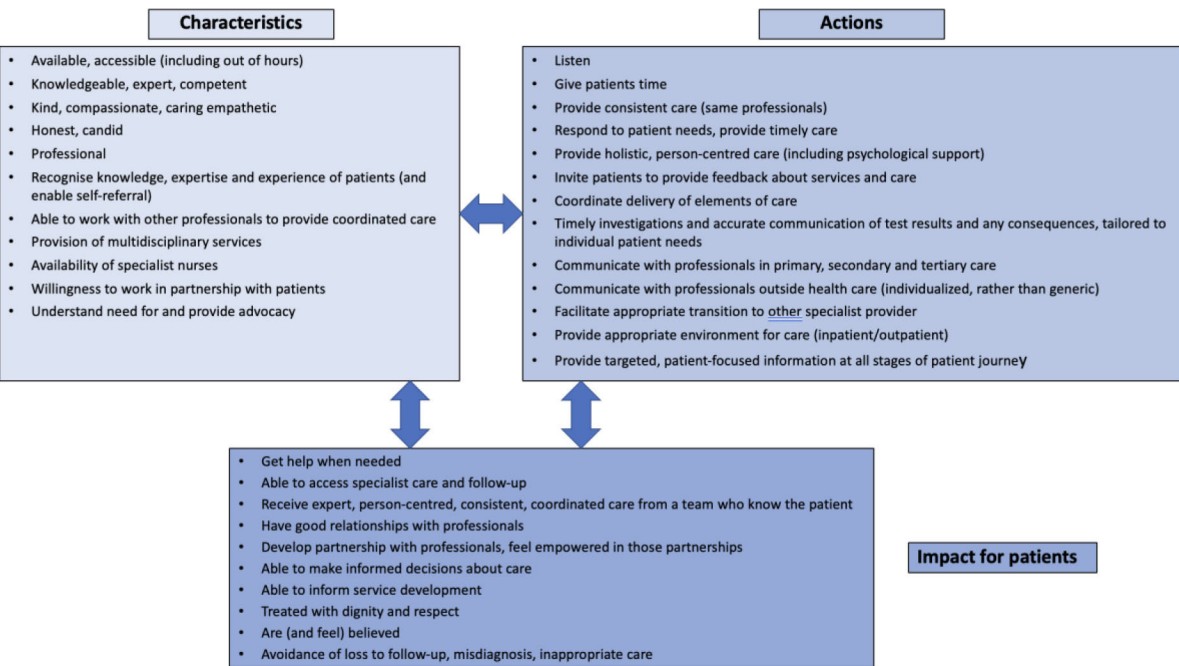

**Figure 2** 'Ideal' characteristics and actions of healthcare services and staff and resulting potential for delivery of 'good' care.

like for people with CHD and their carers (figure 2). This approach recognises that the characteristics of health services and actions of professionals impact how patients (and families) experience and engage with care and our findings resonate with previous studies.[38] People with CHD are a unique group due to their complexity, the requirement for lifelong care and the diversity in how and by whom care is delivered. The NHS Standards for patients with CHD[33] clearly outline what centres should provide in the way of services and access to professionals and this study builds on these by offering insight from the service users' perspective.

## Limitations

Data collection (asynchronous discussion forums on Facebook) facilitated accessibility for participants and resulted in geographically and clinically diverse participants of different ages. However, a broad range of ethnicities was not seen and the vast majority of participants were female, consistent with previous findings that females are more likely to use social media.[39 40] The study design precluded participation from those who could not read English, use social media or were unfamiliar with, or without access to, appropriate technology and potentially those with intellectual, physical and/or sensory disability. Patients and families already in contact with the charities may have been more likely to engage, leading to potential selection bias and limited generalisability of the findings. Due to the format of the data provided, we were unable to link demographic characteristics to responses and it was not always possible to determine how many participants contributed to discussions rather than just signing up to the forum. Inevitably views of more vocal participants may be over-represented although careful reading of the transcript did enable us to determine to some extent posts from the same participant. Finally, some adults with CHD experienced elements of the patient journey some time ago and their experiences may not be reflective of how some of these aspects of care are managed today.

The fourth principle of the NHS Constitution of England[41] states that 'the patient will be at the heart of everything the NHS does' and that 'NHS services must reflect, and should be coordinated around and tailored to, the needs and preferences of patients, their families and their carers.' Delivering good care and a good patient experience are crucial if patients are to engage, and remain engaged, with their healthcare and outcomes are to be optimised[42]; understanding what 'good' care means to patients and families is a key element of that process. Our findings, while indicating that people with CHD share commonalities with other patient groups, also reflect the challenges of delivering comprehensive care to this diverse and growing population. Participants have articulated how they would like care to be delivered and what good care looks like, with an emphasis on individualised, person-centred care delivered within an appropriately resourced, multidisciplinary service. They want to be acknowledged as experts in their own condition and

for the whole life impact of CHD and its treatment to be recognised, not just the physical impact.

## Future directions and conclusion

People with CHD and their parents/carers know how they want care to be delivered and yet, as is evident from the resonance of our findings with those of others, the gap between what patients want (and have a right to expect) and what they receive remains wide. The introduction of standards for the delivery of care to patients with CHD and their families has clearly not resulted in all of those standards being met or in the delivery of equitable care. Patient views need to inform quality improvement initiatives or, recognising that what patients want may not always be deliverable and may remain aspirational, patients' and families' expectations need to be managed. Patients want their voices to be heard and using the results to inform the development of patient-reported experience measures for the CHD population, for routine completion, would enable services to be continually evaluated, benchmarked and improved, with resulting benefits for optimising patients' experiences and outcomes and the opportunity to reduce inequity in health service delivery.

**Acknowledgements** We wish to acknowledge the invaluable contribution to the CHAMPION project of Dr Adrian Chester, a highly respected researcher, committed patient advocate, colleague and friend who sadly died in November 2023. The authors thank The Somerville Foundation, Little Hearts Matter and the Children's Heart Federation for recruiting participants and moderating the forums, and for their contributions, together with the PPI group, to the development and review of the forum questions. Research at Great Ormond Street Hospital is supported by the Great Ormond Street Hospital NIHR Biomedical Research Centre.

**Contributors** JW contributed to the design of the study and undertook the initial analysis of the transcripts; wrote the initial draft of the manuscript and approved the final version. CP contributed to the design of the study; revised the manuscript and approved the final version. LC contributed to the design of the study and undertook the initial analysis of the transcripts; revised the manuscript and approved the final version. AHC contributed to the design of the study and led the study PPI; revised the manuscript and approved the initial submitted version. FK contributed to the design of the study and undertook the initial analysis of the transcripts; revised the manuscript and approved the final version. SC contributed to the design of the study; independently checked that all data related to experience of care were represented appropriately in the themes; revised the manuscript and approved the final version. The guarantor for this manuscript is JW.

**Funding** This study is independent research funded by the National Institute for Health and Care Research (Policy Research Programme, Congenital Heart Audit: Measuring Progress In Outcomes Nationally (CHAMPION), PR-R20-0318-23001).

**Disclaimer** The views expressed in this publication are those of the author(s) and not necessarily those of the NHS, the National Institute for Health Research or the Department of Health and Social Care.

**Competing interests** None declared.

**Patient and public involvement** Patients and/or the public were involved in the design, or conduct, or reporting, or dissemination plans of this research. Refer to the Methods section for further details.

**Patient consent for publication** Not applicable.

**Ethics approval** This study involves human participants and was approved by North of Scotland Research Ethics Service (ref: 20/NS/0022). Participants were not asked to provide informed consent as no patient identifiers were collected by the researchers and the online forums were facilitated by the patient charities. Participant information was provided and consent was assumed if participants signed up and participated in the online discussion forums.

**Provenance and peer review** Not commissioned; externally peer reviewed.

**Data availability statement** All data relevant to the study are included in the article or uploaded as online supplemental information.

**ORCID iDs**
Jo Wray http://orcid.org/0000-0002-4769-1211
Christina Pagel http://orcid.org/0000-0002-2857-1628
Louise Coats http://orcid.org/0000-0003-3422-5497
Sonya Crowe http://orcid.org/0000-0003-1882-5476

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
