## [Reviewer comments · BMJ Open]

ARTICLE DETAILS

TITLE (PROVISIONAL)	What does good care look like to people living with congenital heart disease in the 21st century? Qualitative online, asynchronous discussion forums
AUTHORS	Wray, Jo; Pagel, Christina; Coats, Louise; Chester, Adrian; Kennedy, Fiona; Crowe, Sonya

VERSION 1 – REVIEW

REVIEWER	Milaras, Nikias General Hospital of Athens Ippokrateio, cardiology
REVIEW RETURNED	24-Sep-2023

GENERAL COMMENTS	This is a well written manuscript presenting the point of view of patients suffering from CHD, regarding their medical visits and hospitalization. Patients with CHD are well known to have a much worse health related quality of life when compared to healthy counterparts and are faced with multiple medical visits, hospitalization and multidisciplinary medical examinations throughout their life. That being said, in many if not all health systems, such patients have complaints about the care they receive. This manuscript presents those findings to the reader in a 'narrative' way. Albeit this being a very interesting and serious matter, i didnt find this manuscript original nor very interesting to read.
--

REVIEWER	Sabeti, Fahimeh Iran University of Medical Sciences
REVIEW RETURNED	03-Oct-2023

GENERAL COMMENTS	The manuscript was reviewed. It is a valuable manuscript. Some suggestions are mentioned. Introduction: 1-It is suggested to mention the statistics of congenital heart disease in the world and UK.2-To describe the status of care and follow-up of patients with CHD in UK. Methods: 1-How long were the interview times? How many sessions?2-What were the ethical considerations for the participants?3-Was written informed consent completed?
--

	4-At the end of the quotations, it should be stated who the quotation is from? Adult patient, parent or teenager and carer. Discussion: 1-The discussion will be enriched. Key themes are discussed and results are compared with other studies.
--	--

REVIEWER	Sewell, Taylor Columbia University
REVIEW RETURNED	13-Feb-2024

GENERAL COMMENTS	Thank you for the opportunity to review your work. And thank you for your dedication to ensuring that children with chronic diseases receive care that meets their needs as fully as possible. Your social media-based approach was innovative, as was the way you partnered with disease-specific nonprofit organizations that likely have a great degree of credibility in the eyes of your study population. In the spirit of enhancing your important manuscript, I would recommend giving some additional thought to three principle areas: (1) You summarize your findings by saying: "Participants have articulated how they would like care to be delivered and what good care looks like, with an emphasis on individualised, person-centred care delivered within an appropriately resourced, multidisciplinary service" (Page 13 Line 57 - Page 14 Line 5). This is precisely how I would have imagined this patient population would describe 'good care.' Indeed, it is how I imagine all patients with chronic disease (or other medical complexity) would likely describe "good care." The follow-on then is that I wonder precisely how your research is adding to the literature. Had you hypothesized that CHD patients would have different needs compared to other patient populations (and yet found they were the same)? Are you primarily attempting to call attention to how patients with chronic disease are not receiving care according to NHS standards? Are you primarily trying to provide evidence to administrators looking to revise their patient experience questions for populations living with chronic diseases? I wish your conclusions left me with a clearer sense of how you perceive the additive value of your work. (There actually isn't a Conclusions section, which may be part of the issue.) (2) I had a tough time reading through your six process-of-care themes in the Results section, as their descriptions and examples did not always seem to match their assigned theme. For example: - Within the Relationships theme, you mention "excellent communicators," "confident communication," and "we were listened to"--all of which feel like they may better belong in the Communication theme. - Within the Communication theme, you discuss "making informed decisions," but that was also mentioned in the Relationships theme ("being involved in decision making"). - Within the Access and Coordination theme, you mention "joined-up way of all the hospitals and primary care teams working together", which was also noted in the Communication theme
--

("importance (and often failure) of communication to ensure joined up care")

- Within the Episodes of Care theme, you discuss "Changes to clinic arrangements made as a result of Covid-19...[including] having a video or telephone conversation to discuss those test results"--this feels a lot like Access to me.

I appreciate that the themes naturally overlap in many ways, and some of what I have noted may be unavoidable. However, I would offer that the use of nearly identical words/phrases across themes may be an indication that some edits would be helpful.

(3)

I find Figure 1 to be confusing for three reasons:

(a) Your text's Results section discusses your six process-of-care themes in detail, but these six themes are situated in only one box ('Processes' box) of many boxes in the Figure. The content contained within the Figure's *other* boxes is appropriately noted in the text itself, but that content is all described during the text's presentation of the 'Processes' themes (in the Results section).

Why did you separate out these other boxes in your Figure but still integrate their content into the presentation of your 'Processes' themes in the text? Is this content better conceived as being separate from or part of the 'Processes' themes?

(b) The three larger arrows at the top of the Figure ("Stages of patient journey," "Transition," and "Diagnosis") strike me as a bit odd. The unidirectionality of those three arrows implies continual forward movement in time, yet the boxes located underneath are each present at all of those left-to-right

stages/transitions/diagnostic time points. Why the unidirectional nature of these larger arrows then? (Further, are these three larger arrows adding value to your figure? In my view, they seem to be cluttering more than aiding, especially since the thrust of your manuscript really is on the Process)

(c) I found all the smaller arrows between the boxes distracting. Your manuscript--as I've understood it--is meant to be less a process map than a qualitative compendium. As a result, these arrows distract me from your main purpose more than they add to it.

Additionally, some smaller points for you to consider:

- Page 5 Lines 28-30: "Occasionally...with symptoms": it felt odd that the concluding sentence of the paragraph introduces another concept (which isn't itself entirely necessary to state given the topic of your manuscript).

- Page 6 Line 21: "patient co-researcher": what does this mean? Does it mean that author AC is a patient?

- Page 6 Line 44: "parents/carers of younger children": how did you define 'younger children'?

- Page 6 Line 44 / Line 52: "parents/carers of younger children" / "parents/carers of young people": what about parents/carers of adolescents? Parents/carers of adults?

- Page 6 Line 46: "CHD severity (complex/less complex)": Table 2 defines CHD severity more specifically: single ventricle condition

vs biventricular condition vs unknown. I'd recommend harmonizing these two references to CHD severity.

- Page 6 Lines 46-48: "The research team...charity collaborators": this feels more suited for the above PPI sub-section.
- Page 6 Line 52: "parents/carers of younger people": how did you define 'young people'? And is this different than what you meant by "younger children" (Line 44)?
- Page 6 Lines 52-55: "Carers of adult patients...discussion groups": these carers could participate in the adult patient groups and also the parent/carer groups? Meaning they had a choice as to which kind of group to participate in?
- Methods section: in what months/years was the study conducted?
- Methods section: I would be interested to see an Appendix that lists all the questions you asked across your forums (as well as, if possible, the questions that "charities could introduce" (Page 7 Line 5).
- Methods section: was there an enrollment period before the questions began to be posed to participants? Or was enrollment ongoing? If ongoing, could participants see (and respond to) old questions (or only those posted after they enrolled)?
- Page 7 Line 23: "independently by members": how many members?
- Page 7 Lines 28-32: "The themes and transcripts...in the themes": did FK make any changes to the themes?
- Page 8 Line 16: "consultant": I presume this is British English for what we typically call a "specialist" in American English? Perhaps consider asking the journal editor whether it would be helpful to define 'consultant' for your international audience (e.g., "have a 'great relationship with [the] consultant [i.e., specialist]...").
- Page 8 Line 46: "with the need for others to advocate for them": I can't tell if "others" specifically (and solely) means CNS nurses here, or if it also includes other staff, family members, etc.
- Page 9 Line 14: "communication both with and between professionals": this paragraph exclusively discusses communication *with* professionals, not communication *between* professionals. I appreciate this is the topic sentence of the entire Communication sub-section, and I know that later in the sub-section you discuss communication between professionals. However, I wonder if it would read a little cleaner if you left Line 14 reading "communication with professionals" and then had a new topic sentence at the start of the next paragraph (Line 30) where you introduce the idea of communication *between* professionals?
- Page 9 Line 16: "functional elements (accuracy, timeliness), content": I'm curious why you viewed accuracy of communication to be a functional element instead of being related to content.
- Page 10 Line 14: "Access and coordination" sub-section header: I'm wondering why you combined these two key themes for this sub-section when you otherwise have kept them as separate and distinct when you list the six themes (e.g., Page 7 Lines 52-55). If 'Access' and 'Coordination' really do need to be discussed together in this single sub-section, it makes me wonder whether they truly are two distinct themes or whether there is a single theme of 'access and coordination' (such that you have five key themes instead of six)?
- Page 11 Line 3: "everything under one roof not non-guch [ACHD] hosp": as written, I can't tell if it's 'guch' or 'non-guch' that means

'ACHD'. That is, would this sentence otherwise read as 'everything under one roof not non-ACHD hosp' or 'everything under one roof not ACHD hosp'?

- Page 11 Line 19: "Episodes of care" sub-section header: I would consider changing the title of this theme--both here and elsewhere when you list it--to "Experience of discrete episodes of care" (or something similar). "Episodes of care" is quite broad and encompasses many more elements than those you highlight, which all seem to be about the experience of those discrete episodes. As such, I think finding a way to narrow your theme label might help better set the reader's expectations of what you're going to discuss.

- Page 11 Line 52: "Holistic care" sub-section header: is there a reason you don't simply call this theme--both here and elsewhere when you list it--"Psychological support"? That feels closer to what you actually discuss in this sub-section.

- Results section: Where was the description of the sub-themes that you noted on Page 7 Line 25?

- You reference "outcomes" a number of times in the manuscript (e.g., Page 7 Line 57, Page 12 Line 37). I presume you mean 'health outcomes' (as opposed to, say, 'experience of care outcomes'). If you *do* mean health outcomes, I would recommend you review your Results section, as I saw only minimal mention of what truly reads as health outcomes to me (e.g., "I had a stroke which I believe could have been avoided" on Page 10 Line 26). There were other allusions to health outcomes (e.g., "coming up with a plan" on Page 9 Line 39), but the mention of the stroke was the only real health outcomes reference I could readily identify.

- Page 13 Line 19: "supporting previous findings": this language almost suggests that one of your study's aims was to support or refute these previous findings. Consider whether "consistent with previous findings" may be more appropriate.

- Page 13 Line 25: "intellectual disability": what about physical disability (e.g., blindness, limited dexterity)?

- Table 2: 'Participants' box: after you consider my suggestions above regarding the language used to describe your participant categories in the Methods section, I would recommend utilizing that same Methods language here in Table 2.

- Table 2: 'Participate age group' box: ">61 years": did you mean ">60 years"? For the 'Age group of person with CHD' box, you write "16-18 years" and ">18 years". Perhaps it would be less fraught if, for both boxes, you said "61+ years" and "19+ years", respectively?

- Figure 2: 'Characteristics' box: "Provision of multidisciplinary services": consider whether this is best construed as a characteristic or an action. (It would read as "Provide multidisciplinary services" if an action).

- Figure 2: 'Actions' box: "Responsiveness to patient needs, provision of timely care": consider whether this could have a greater action orientation if re-phrased as "Respond to patient needs, provide timely care".

- Figure 2: 'Impact for patients' box: "Develop partnership with professionals, feel empowered": feel empowered about what exactly?

- Figure 2: double-headed arrows: why are there two of them (instead of just one)? And given their placement, it visually seems

	that you're suggesting only 'Actions' and 'Impact for patients' are interrelated (leaving 'Characteristics' unrelated). From context, I don't think this is your intention, so perhaps consider a different layout or visual indicator of the interconnectedness of these three boxes.
--	--

REVIEWER	Ziniel, Sonja University of Colorado at Denver - Anschutz Medical Campus
REVIEW RETURNED	21-Feb-2024

GENERAL COMMENTS	This manuscript presents a qualitative study on the views of patients with congenital heart disease (CHD) and/or their parents/carers about their experiences with health care and what “good” care looked like. The authors used five asynchronous, online discussion groups via the Facebook pages of three patient charities to gather the qualitative data. The transcripts from these online discussion groups were thematically analyzed with six themes emerging: relationships, communication, access, coordination, episodes of care, and holistic care. Not surprisingly, patients describe good care as care that is person-centered, coordinated and integrated among all services, easily accessible, and with clinicians recognizing patients as experts in their disease. The authors nicely visualize the themes and the relationships between them in Figure 1. My main concerns regarding this manuscript relate to the discussion of the results. First, the discussion is very short and introduces a new Figure 2 which I would have expected to be presented in the Result section rather than in the Discussion. Second, while this study is contextually embedded in the priorities identified in the NHS Outcomes Framework, these themes, and their contribution to “good” care (irrelevant of the disease) are well known in the literature around patient satisfaction, and care integration, specifically for population with complex diseases or chronic illnesses. The authors, however, fail to compare or contrast their findings to what has been previously found in this research area regarding CHD on the one hand and patient satisfaction factors on the other hand. While CHD/ACHD has some specific characteristics that the authors mentioned in the background, it is important to discuss, how much it has in common with other chronic/complex diseases regarding a “good” care experience. Besides these main concerns, there are a few smaller thoughts:  1) Methods: Please note how many of the forums were for adult patients, compared to adolescent patients and parents/carers of young people with CHD 2) Results:  a. Please note how many participants participated overall and in each type of forum. You indicated it in the abstract, but it is missing here. b. Please provide the total number of respondents for Table 2 c. Were there any themes that were highlighted more in one forum comp 3) Discussion:  a. I would rephrase the first sentence, line 35 as follows to be consistent: ... identifies six key themes (relationships,
--

	communication, access, coordination, episodes of care, and holistic care)... b. Limitations: The limitations are well stated. However, I am wondering if the authors can determine how many individuals actually contributed to the asynchronous discussion groups (compared to just signing up). While in a focus group, for example, moderators can encourage more quiet participants to speak up, this seems hard to do in an asynchronous group and therefore I would expect the views of more vocal individuals to be overrepresented in the comments. This should be at least also acknowledged in the limitations.
--	---

VERSION 1 – AUTHOR RESPONSE

Reviewer: 1

Dr. Nikias Milaras, General Hospital of Athens Ippokrateio

Comments to the Author:

This is a well written manuscript presenting the point of view of patients suffering from CHD, regarding their medical visits and hospitalization.

Patients with CHD are well known to have a much worse health related quality of life when compared to healthy counterparts and are faced with multiple medical visits, hospitalization and multidisciplinary medical examinations throughout their life.

That being said, in many if not all health systems, such patients have complaints about the care they receive.

This manuscript presents those findings to the reader in a 'narrative' way.

Albeit this being a very interesting and serious matter, i didnt find this manuscript original nor very interesting to read.

We hope that the revisions to our paper have made it more interesting for you to read.

Reviewer: 2

Dr. Fahimeh Sabeti, Iran University of Medical Sciences

Comments to the Author:

The manuscript was reviewed. It is a valuable manuscript. Some suggestions are mentioned.

Introduction:

1-It is suggested to mention the statistics of congenital heart disease in the world and UK.

We have now included the incidence of CHD.

2-To describe the status of care and follow-up of patients with CHD in UK.

We have clarified that the approach to follow-up of patients in the UK is as described in the first paragraph of the background.

Methods:

1-How long were the interview times? How many sessions?

We did not undertake interviews with patients as data were collected via online, asynchronous discussion forums. There were no set sessions. Participants could answer as many questions as they wished during the time that the forum was live (12-24 weeks).

2-What were the ethical considerations for the participants?

We have added further information about how the online discussion forums were moderated and the use of a standard operating procedure which included information about processes for managing inappropriate and/or offensive messaging and distressed users.

3-Was written informed consent completed?

Written informed consent was not requested – providing information was taken as consent to participate, as is frequently the case with anonymous online data collection. Participants were given information about study, what participation in the online forums would involve and how the data would be used.

4-At the end of the quotations, it should be stated who the quotation is from? Adult patient, parent or teenager and carer.

Thank you – we have made it clear that participant quotes are from adult patients unless otherwise stated (please see paragraph 2, results)

Discussion:

1-The discussion will be enriched. Key themes are discussed and results are compared with other studies.

We have added further discussion of the key themes in the context of the wider literature.

Reviewer: 3

Dr. Taylor Sewell, Columbia University

Comments to the Author:

Thank you for the opportunity to review your work. And thank you for your dedication to ensuring that children with chronic diseases receive care that meets their needs as fully as possible. Your social media-based approach was innovative, as was the way you partnered with disease-specific nonprofit organizations that likely have a great degree of credibility in the eyes of your study population. In the spirit of enhancing your important manuscript, I would recommend giving some additional thought to three principle areas:

(1)

You summarize your findings by saying: “Participants have articulated how they would like care to be delivered and what good care looks like, with an emphasis on individualised, person-centred care delivered within an appropriately resourced, multidisciplinary service” (Page 13 Line 57 — Page 14 Line 5). This is precisely how I would have imagined this patient population would describe ‘good care.’ Indeed, it is how I imagine all patients with chronic disease (or other medical complexity) would likely describe “good care.” The follow-on then is that I wonder precisely how your research is adding to the literature. Had you hypothesized that CHD patients would have different needs compared to other patient populations (and yet found they were the same)? Are you primarily attempting to call attention to how patients with chronic disease are not receiving care according to NHS standards? Are you primarily trying to provide evidence to administrators looking to revise their patient experience questions for populations living with chronic diseases? I wish your conclusions left me with a clearer sense of how you perceive the additive value of your work. (There actually isn’t a Conclusions section, which may be part of the issue.)

We agree that one might expect patients with chronic conditions to describe good care in the way in which they did. Our aim was to understand current experiences of care and how that related to how patients would like care to be delivered and, as part of the wider project, outcomes evaluated. Patients with CHD are a unique group in terms of how their care is delivered in the UK and their need for specialist life-long follow-up and understanding whether the NHS standards for the delivery of care to patients with CHD are perceived to be being met was also important. We have added further text to the discussion and a conclusion to make this clearer.

(2)

I had a tough time reading through your six process-of-care themes in the Results section, as their descriptions and examples did not always seem to match their assigned theme. For example:

- Within the Relationships theme, you mention “excellent communicators,” “confident communication,” and “we were listened to”—all of which feel like they may better belong in the Communication theme.
- Within the Communication theme, you discuss “making informed decisions,” but that was also

mentioned in the Relationships theme (“being involved in decision making”).

- Within the Access and Coordination theme, you mention “joined-up way of all the hospitals and primary care teams working together”, which was also noted in the Communication theme (“importance (and often failure) of communication to ensure joined up care”)

- Within the Episodes of Care theme, you discuss “Changes to clinic arrangements made as a result of Covid-19...[including] having a video or telephone conversation to discuss those test results”—this feels a lot like Access to me.

Thank you for highlighting these issues. There is overlap between themes, as the reviewer notes, but we have addressed the examples identified and made it clearer where the quotes fit.

I appreciate that the themes naturally overlap in many ways, and some of what I have noted may be unavoidable. However, I would offer that the use of nearly identical words/phrases across themes may be an indication that some edits would be helpful.

Thank you. We recognise the valid points being made by the reviewer and have revised our themes accordingly, simplifying them to four rather than six to avoid some of the overlap. The themes are all interlinked, some more closely than others, but we hope that the revisions provide more clarity and distinction between the themes. As the reviewer correctly identified, some of the quotes cut across two themes so we have revised the text so that quotes fit more clearly into one rather than two themes.

(3)

I find Figure 1 to be confusing for three reasons:

(a) Your text’s Results section discusses your six process-of-care themes in detail, but these six themes are situated in only one box (‘Processes’ box) of many boxes in the Figure. The content contained within the Figure’s *other* boxes is appropriately noted in the text itself, but that content is all described during the text’s presentation of the ‘Processes’ themes (in the Results section). Why did you separate out these other boxes in your Figure but still integrate their content into the presentation of your ‘Processes’ themes in the text? Is this content better conceived as being separate from or part of the ‘Processes’ themes?

(b) The three larger arrows at the top of the Figure (“Stages of patient journey,” “Transition,” and “Diagnosis”) strike me as a bit odd. The unidirectionality of those three arrows implies continual forward movement in time, yet the boxes located underneath are each present at all of those left-to-right stages/transitions/diagnostic time points. Why the unidirectional nature of these larger arrows then? (Further, are these three larger arrows adding value to your figure? In my view, they seem to be cluttering more than aiding, especially since the thrust of your manuscript really is on the Process)

(c) I found all the smaller arrows between the boxes distracting. Your manuscript—as I’ve understood it—is meant to be less a process map than a qualitative compendium. As a result, these arrows distract me from your main purpose more than they add to it.

We have revised Figure 1 and simplified it to try and better illustrate that the processes of care happened in a range of settings, against a backdrop of different contextual factors (stage of patient journey, patient characteristics) with a resulting impact on patient experience and health outcomes. We appreciate the detailed feedback on the figure and hope that we have addressed the concerns of the reviewer.

Additionally, some smaller points for you to consider:

- Page 5 Lines 28-30: “Occasionally...with symptoms”: it felt odd that the concluding sentence of the paragraph introduces another concept (which isn’t itself entirely necessary to state given the topic of your manuscript).

We have left this sentence in as late diagnosis (in adulthood) is something that some participants mentioned and had experienced. We mention it here because this group do not fit the chronology of follow-up that we describe for most patients with CHD that we describe earlier in this paragraph.

- Page 6 Line 21: “patient co-researcher”: what does this mean? Does it mean that author AC is a patient?

Yes, AC is a patient. We have reworded this to make it clearer.

- Page 6 Line 44: “parents/carers of younger children”: how did you define ‘younger children’?

We have added our definition of younger children as <12 years.

- Page 6 Line 44 / Line 52: “parents/carers of younger children” / “parents/carers of young people”: what about parents/carers of adolescents? Parents/carers of adults?

Thank you for highlighting this oversight – we have made it clear that we also included parents/carers of adolescent and adult patients.

- Page 6 Line 46: “CHD severity (complex/less complex)”: Table 2 defines CHD severity more specifically: single ventricle condition vs biventricular condition vs unknown. I’d recommend harmonizing these two references to CHD severity.

We have added single and biventricular conditions to the text.

- Page 6 Lines 46-48: “The research team...charity collaborators”: this feels more suited for the above PPI sub-section.

We have left this sentence in the data collection section. The PPI section refers specifically to our PPI group and their involvement whereas the charity collaborators were directly involved in data collection.

- Page 6 Line 52: “parents/carers of younger people”: how did you define ‘young people’? And is this different than what you meant by “younger children” (Line 44)?

We have reworded this to make it clear that we mean parents of children and adolescents.

- Page 6 Lines 52-55: “Carers of adult patients...discussion groups”: these carers could participate in the adult patient groups and also the parent/carer groups? Meaning they had a choice as to which kind of group to participate in?

This was not clear so we have clarified that carers of adult patients joined the adult patient groups. They did not have a choice.

- Methods section: in what months/years was the study conducted?

We have added this information.

- Methods section: I would be interested to see an Appendix that lists all the questions you asked across your forums (as well as, if possible, the questions that “charities could introduce” (Page 7 Line 5).

We have added to the example of questions provided in Table 1 for one of the forums. We have not provided all of the questions as some of them are not relevant for the data reported in this manuscript.

- Methods section: was there an Enrolment period before the questions began to be posed to participants? Or was Enrolment ongoing? If ongoing, could participants see (and respond to) old questions (or only those posted after they enrolled)?

Enrolment was ongoing so participants could respond to questions posted earlier. This detail has been added to the text.

- Page 7 Line 23: “independently by members”: how many members?

We have clarified that three members of the team independently coded the transcript.

- Page 7 Lines 28-32: “The themes and transcripts...in the themes”: did FK make any changes to the themes?

No changes were made to the themes by SC (not FK as incorrectly stated, FK coded the transcript)

- Page 8 Line 16: “consultant”: I presume this is British English for what we typically call a “specialist” in American English? Perhaps consider asking the journal editor whether it would be helpful to define ‘consultant’ for your international audience (e.g., “have a ‘great relationship with [the] consultant [i.e., specialist]...”).

We will defer to the editor on this point as it is our understanding that there are a number of possible terms such as attending as well as specialist.

- Page 8 Line 46: “with the need for others to advocate for them”: I can’t tell if “others” specifically (and solely) means CNS nurses here, or if it also includes other staff, family members, etc.

We have clarified that this means staff and/or family members.

- Page 9 Line 14: “communication both with and between professionals”: this paragraph exclusively discusses communication *with* professionals, not communication *between* professionals. I appreciate this is the topic sentence of the entire Communication sub-section, and I know that later in the sub-section you discuss communication between professionals. However, I wonder if it would read a little cleaner if you left Line 14 reading “communication with professionals” and then had a new topic sentence at the start of the next paragraph (Line 30) where you introduce the idea of communication *between* professionals?

Thank you, we have revised the text as suggested.

- Page 9 Line 16: “functional elements (accuracy, timeliness), content”: I’m curious why you viewed accuracy of communication to be a functional element instead of being related to content.

We agree with the reviewer and have revised the text.

- Page 10 Line 14: “Access and coordination” sub-section header: I’m wondering why you combined these two key themes for this sub-section when you otherwise have kept them as separate and distinct when you list the six themes (e.g., Page 7 Lines 52-55). If ‘Access’ and ‘Coordination’ really do need to be discussed together in this single sub-section, it makes me wonder whether they truly are two distinct themes or whether there is a single theme of ‘access and coordination’ (such that you have five key themes instead of six)?

Please see above. We have simplified our themes such that we now have four themes with access and coordination being seen as one theme (because of the degree of overlap between the two concepts).

- Page 11 Line 3: “everything under one roof not non-guch [ACHD] hosp”: as written, I can’t tell if it’s ‘guch’ or ‘non-guch’ that means ‘ACHD’. That is, would this sentence otherwise read as ‘everything under one roof not non-ACHD hosp’ or ‘everything under one roof not ACHD hosp’

We have added an explanation of GUCH (now called ACHD) at the end of the paragraph to clarify this.

- Page 11 Line 19: “Episodes of care” sub-section header: I would consider changing the title of this theme—both here and elsewhere when you list it—to “Experience of discrete episodes of care” (or something similar). “Episodes of care” is quite broad and encompasses many more elements than those you highlight, which all seem to be about the experience of those discrete episodes. As such, I think finding a way to narrow your theme label might help better set the reader’s expectations of what you’re going to discuss.

We agree with this suggestion, thank you.

- Page 11 Line 52: “Holistic care” sub-section header: is there a reason you don’t simply call this theme—both here and elsewhere when you list it—“Psychological support”? That feels closer to what you actually discuss in this sub-section.

We have renamed this theme as suggested.

- Results section: Where was the description of the sub-themes that you noted on Page 7 Line 25?

Thank you for pointing this out – this should not have been noted and reference to subthemes has been removed.

- You reference “outcomes” a number of times in the manuscript (e.g., Page 7 Line 57, Page 12 Line 37). I presume you mean ‘health outcomes’ (as opposed to, say, ‘experience of care outcomes’). If you *do* mean health outcomes, I would recommend you review your Results section, as I saw only minimal mention of what truly reads as health outcomes to me (e.g., “I had a stroke which I believe

could have been avoided" on Page 10 Line 26). There were other allusions to health outcomes (e.g., "coming up with a plan" on Page 9 Line 39), but the mention of the stroke was the only real health outcomes reference I could readily identify.

The reviewer is correct in assuming that by outcomes we mean health outcomes and we have made that clearer and added some further examples in the results section where participants describe health outcomes.

- Page 13 Line 19: "supporting previous findings": this language almost suggests that one of your study's aims was to support or refute these previous findings. Consider whether "consistent with previous findings" may be more appropriate.

We have amended the text as suggested.

- Page 13 Line 25: "intellectual disability": what about physical disability (e.g., blindness, limited dexterity)?

Thank you for highlighting the omission of physical disabilities – we have added this.

- Table 2: 'Participants' box: after you consider my suggestions above regarding the language used to describe your participant categories in the Methods section, I would recommend utilizing that same Methods language here in Table 2.

We have used single ventricle and biventricular conditions in the text and table.

- Table 2: 'Participate age group' box: ">61 years": did you mean ">60 years"? For the 'Age group of person with CHD' box, you write "16-18 years" and ">18 years". Perhaps it would be less fraught if, for both boxes, you said "61+ years" and "19+ years", respectively?

Thank you – we have amended as suggested.

- Figure 2: 'Characteristics' box: "Provision of multidisciplinary services": consider whether this is best construed as a characteristic or an action. (It would read as "Provide multidisciplinary services" if an action).

We have left this as a characteristic although agree it could also be construed as an action.

- Figure 2: 'Actions' box: "Responsiveness to patient needs, provision of timely care": consider whether this could have a greater action orientation if re-phrased as "Respond to patient needs, provide timely care".

Thank you, we agree with this suggestion and have revised the text.

- Figure 2: 'Impact for patients' box: "Develop partnership with professionals, feel empowered": feel empowered about what exactly?

We have added text to state that it is about feeling empowered in the partnerships with professionals.

- Figure 2: double-headed arrows: why are there two of them (instead of just one)? And given their placement, it visually seems that you're suggesting only 'Actions' and 'Impact for patients' are interrelated (leaving 'Characteristics' unrelated). From context, I don't think this is your intention, so perhaps consider a different layout or visual indicator of the interconnectedness of these three boxes.

The reviewer is correct and we have revised the diagram to illustrate the interconnectedness of the three boxes.

Reviewer: 4

Dr. Sonja Ziniel, University of Colorado at Denver - Anschutz Medical Campus

Comments to the Author:

This manuscript presents a qualitative study on the views of patients with congenital heart disease (CHD) and/or their parents/carers about their experiences with health care and what "good" care looked like. The authors used five asynchronous, online discussion groups via the Facebook pages of three patient charities to gather the qualitative data. The transcripts from these online discussion groups were thematically analyzed with six themes emerging: relationships, communication, access, coordination, episodes of care, and holistic care. Not surprisingly, patients describe good care as care that is person-centered, coordinated and integrated among all services, easily accessible, and with

clinicians recognizing patients as experts in their disease. The authors nicely visualize the themes and the relationships between them in Figure 1.

My main concerns regarding this manuscript relate to the discussion of the results. First, the discussion is very short and introduces a new Figure 2 which I would have expected to be presented in the Result section rather than in the Discussion.

We have added further detail into the discussion. We have left the introduction of Figure 2 in the discussion as we built on the conceptual map of Entwistle to develop this figure and do not feel that discussion of that is appropriate in the results section of the manuscript.

Second, while this study is contextually embedded in the priorities identified in the NHS Outcomes Framework, these themes, and their contribution to “good” care (irrelevant of the disease) are well known in the literature around patient satisfaction, and care integration, specifically for population with complex diseases or chronic illnesses. The authors, however, fail to compare or contrast their findings to what has been previously found in this research area regarding CHD on the one hand and patient satisfaction factors on the other hand. While CHD/ACHD has some specific characteristics that the authors mentioned in the background, it is important to discuss, how much it has in common with other chronic/complex diseases regarding a “good” care experience.

Thank you – we have added further text to the discussion to provide more context in terms of the wider literature.

Besides these main concerns, there are a few smaller thoughts:

1) Methods: Please note how many of the forums were for adult patients, compared to adolescent patients and parents/carers of young people with CHD

This information has been added.

2) Results:

a. Please note how many participants participated overall and in each type of forum. You indicated it in the abstract, but it is missing here.

We have included the total number of participants in the results section; the number of participants in each type of forum are provided in Table 2.

b. Please provide the total number of respondents for Table 2

We have added this to the title of Table 2.

c. Were there any themes that were highlighted more in one forum compared

The parent/carer and adult patient forums were broadly similar in the themes that were discussed. In the young people’s forum, where numbers were much smaller (n=11), there was less discussion of experience of surgery or complications. We have mentioned this in the discussion.

3) Discussion:

a. I would rephrase the first sentence, line 35 as follows to be consistent: ... identifies six key themes (relationships, communication, access, coordination, episodes of care, and holistic care)...

The themes have been simplified and this sentence now reflects the fact that there were four key themes and what they were.

b. Limitations: The limitations are well stated. However, I am wondering if the authors can determine how many individuals actually contributed to the asynchronous discussion groups (compared to just signing up). While in a focus group, for example, moderators can encourage more quiet participants to speak up, this seems hard to do in an asynchronous group and therefore I would expect the views of more vocal individuals to be overrepresented in the comments. This should be at least also acknowledged in the limitations.

Thank you – we have added this to the limitations as we were not always able to tell how many people contributed, due in part to how the charities shared the data with us.

We believe that the revisions have strengthened our manuscript and we hope that it will now be considered acceptable for publication in *BMJ Open*.

VERSION 2 – REVIEW

REVIEWER	Sewell, Taylor Columbia University
REVIEW RETURNED	29-Apr-2024

GENERAL COMMENTS	Thank you for your thorough revision; I find this iteration of the manuscript to be meaningfully clearer. Through both the edits you've made to your manuscript and the responses you wrote to my initial review, you have addressed all of my initial concerns with one exception: I still struggle to understand Figure 1. The entirety of your manuscript reflects the "Four linked themes related to processes of care" (Page 8 Line 14) that you identified. This is clearly articulated in Figure 1's red box (titled 'Processes of care'). But I still don't understand the blue 'Context' and 'Setting' boxes in Figure 1. There are no sections in your manuscript dedicated to 'Context' or 'Setting'; rather, your text folds these elements into your analysis of (and examples of) the four process-of-care themes. It remains confusing to me, then, that you give equivalent space and visual weight to these blue 'Context' and 'Setting' boxes in Figure 1. Similarly, your text presents this notion of 'context' as a finding separate from your four process-of-care themes (on Page 8 Lines 23-26 with "In addition...were described"). Beyond that sentence, though, the text doesn't appear to further articulate this finding as distinct from the four process-of-care themes; rather, as noted above, you fold these 'context' elements into your analysis of (and examples of) the four themes. (The same is true for the way you note the 'context' in your abstract (Page 3 Lines 50-55: "In addition...own condition").) Beyond that, below you'll find a few less critical thoughts. Thank you again for your work, your commitment to children's health, and the opportunity to review. -----  - Page 3 Line 19: "closed": would "controlled" be more accurate than "closed"? - Page 5 Line 37: "A&E": please define this abbreviation - Page 6 Lines 30-32: "(comprising three...with CHD)": as written, this parenthetical sounds like it pertains to the immediately preceding "overall study," but from context, I think you meant it to modify the "PPI group" instead. Consider re-situating this phrase so it's a bit clearer what it refers to. - Page 6 Line 34: "on the findings": of the present manuscript? - Page 6 Line 37: "prior to submission": of the present manuscript? - Page 8 Line 14: "Four linked...were identified": this almost sounds like you identified four themes related to processes of care, plus some additional themes related to another topic. Consider instead: "Four linked themes were identified, all related to processes of care"
--

	 - Page 9 Line 41: "we know when we feel sh*tty": consider whether the curse word should be partially redacted (e.g., with asterisks). - Page 9 Line 50: "cardiologist with 'an interest in GUCH' decided": you subsequently define 'GUCH' via a parenthetical on Page 12 Lines 19-21, but I would recommended moving that parenthetical forward to this first mention of "GUCH". (This will also make it clear what "ACHD" means when you use that abbreviation for the first time on Page 10 Line 25.) - Page 10 Lines 3-5: "results from tests...to GP": this quote seems to demonstrate the 'mechanisms of communication' element of the 'practical aspects of communication' under discussion in this sentence. Had you intended to also provide a quote that demonstrates the 'functional' and 'content' components as well? Since you list 'functional,' 'content,' and 'mechanisms' all together in the same sentence, it feels odd that your representative quote seems to only address one of these three. - Page 10 Line 25: "EP": please define this abbreviation - Page 15 Lines 48-50: "although care reading...same participant": I find myself wondering 'to what effect?' You were able to ascertain that some posts came from the same person, but that doesn't seem to meaningfully modify the first half of this sentence, nor is there a follow-on about, say, the frequency with which you noted these repeat posters. So I just don't fully understand the point you're trying to make by including this clause. - Page 16 Line 32: "Future directions and conclusion": was this meant to be bolded (and at the same structural level as, say, the "Discussion" section)? - Table 2: "Participants" cell: "Young people with CHD": does 'young' here mean <12 years or <18 years? - Table 2: "Participants" cell: "Parents/carers of children with CHD": does 'children here mean <12 years or <18 years? - Table 2: "Location of home" cell: Does this refer to the location of the patient or the location of the participant in the forum?
--	---

REVIEWER	Ziniel, Sonja University of Colorado at Denver - Anschutz Medical Campus
REVIEW RETURNED	17-May-2024

GENERAL COMMENTS	Thank you for your detailed revisions to this manuscript. My previous concerns have been addressed.
---

VERSION 2 – AUTHOR RESPONSE